# Changes in tooth brushing frequency and its associated factors from 2006 to 2014 among French adolescents: Results from three repeated cross sectional HBSC studies

Gabriel Fernandez de Grado[1,2,3,4]*, Virginie Ehlinger[4], Emmanuelle Godeau[4,5], Catherine Arnaud[4], Cathy Nabet[4,6], Nadia Benkirane-Jessel[1,2], Anne-Marie Musset[1,2,3], Damien Offner[1,2,3]

1 INSERM (French National Institute of Health and Medical Research), "Regenerative Nanomedicine" laboratory, UMR 1260, Faculté de Médecine, FMTS, Strasbourg, France, 2 Université de Strasbourg, Faculté de Chirurgie Dentaire, Strasbourg, France, 3 Hôpitaux Universitaires de Strasbourg, Strasbourg, France, 4 UMR 1027 INSERM, Paul Sabatier University, Toulouse, France, 5 EHESP school of public health, Rennes, France, 6 Department of Epidemiology and Public Health, Paul Sabatier University, Toulouse University Hospital, Toulouse, France

* gabfdg@free.fr

## Abstract

### Objectives

This study aimed to evaluate in the changes in the percentage of adolescents who brush their teeth twice a day and the association with socio-economic status and health behaviors between 2006, 2010 and 2014 among adolescents from the French cross-sectional studies of the Health Behavior in School-aged Children (HBSC) survey.

### Methods

Our sample included 18727 adolescents aged 11, 13 or 15 years old (y/o). The relationship between toothbrushing frequency (TBF) and eating habits, health and socio-economic status markers, family status, school perception, substance use, sedentary lifestyle and physical activity, together with their evolution over the 3 studies, were investigated using multivariate logistic regression.

### Results

The proportion of adolescents brushing twice a day increased from 68.8% in 2006 to 70.8% in 2010 and 78.8% in 2014 (p<0.0001). Notable associated factors (p<0.0001) were: being a girl (adjusted Odds Ratio = 1.5) and, even more, an older girl (aOR 1.5 for 15 y/o vs 11 y/o girls), having breakfast (aOR 1.4) and eating fruits daily (aOR 1.6), excellent perceived health (aOR 1.2), obesity or overweight (aOR 0.6), being bullied at school (aOR 0.8), and perceived family wealth (aOR 1.4 for High vs Low). No impact from any associated factor changed over the 3 studies.

**Data Availability Statement:** Data are fully available on: https://www.uib.no/en/hbscdata/113290/open-access.

**Funding:** The author(s) received no specific funding for this work.

**Competing interests:** The authors have declared that no competing interests exist.

## Conclusions

Among French adolescents, TBF improved from 2006 to 2014. TBF was significantly associated with other health behaviors. These associations stayed similar in 2006, 2010 and 2010. This increase in TBF may be linked with global prevention programs developed during this time period. These programs should be maintained and associated with more specific ones targeting and adapted to disadvantaged populations, in order to reduce inequalities in oral hygiene and oral health.

## Introduction

Oral diseases such as dental caries and periodontal diseases are among the most frequent chronic diseases around the world [1]. Dental caries is the most frequent childhood disease and the vast majority of adults in developed countries, if not worldwide, suffer from at least one oral pathology [1, 2]. Among 12 years old (y/o) French children, the DMFT index (Decayed, Missing or Filled Teeth) decreased from 4.20 to 1.23 between 1987 and 2006, while the proportion of children free of dental caries increased from 12% to 55.9% over the same period [3, 4]. Despite these encouraging observations, around half of the children still suffered from carious diseases.

The last study on the oral status of (12 year old) French children described an improvement in the proportion of children free from dental caries, from 56% in 2006 to 66% in 2010, among children participating in a national voluntary prevention program with an attendance of only 33% [4, 5]. Thus, it is likely that the oral status of the population, qualified as good overall, is overestimated in this study. Another study, in French schools in 2009, showed only 56% of adolescents to be caries-free, a proportion around 10% lower than the 2010 sample previously described [6].

Oral hygiene, whose main element is toothbrushing, is an efficient, low-cost and easy-to-implement method to prevent almost all oral diseases, especially caries and gingivitis [7]. A recent meta-analysis confirmed that an increase in tooth brushing frequency is associated with a lower incidence and increment of carious lesions [8]. There is overall national and international consensus to advise a twice a day frequency of tooth brushing to prevent oral diseases [7, 9, 10]. It has also been shown that brushing more than once a day before the age of 12 leads to a stable tooth brushing frequency during the following years [11]. Tooth brushing is not only a preventive act; its frequency also provides an easy-to-assess indicator of oral health habits.

While necessary to improve oral health, changes in dietary habits and tooth brushing frequency and efficiency are hard to implement [9, 12]. A low socio-economic status (SES) during childhood is associated with a lower tooth brushing frequency and a higher caries prevalence among adults, as are other aggravating factors for oral health such as smoking or unhealthy diet [13–15]. Upward changes in SES in adulthood are not sufficient to recover from these adverse effects [16].

A previous work on data from the 2010 French HBSC study (Health Behavior in School-aged Children) highlighted the strong relationships between tooth brushing frequency, health behaviors (mostly dietary), health and body perception, and environmental socio-economic factors [17].

We wanted to evaluate the trends in toothbrushing frequency over successive studies, as well as the evolution of the relationships between toothbrushing frequency and previously identified associated factors. Our main hypothesis was that adolescent's behavior regarding toothbrushing frequency may have improved. Our secondary question was "Among

adolescents, what is the effect of associated factors (Diet, Health and body perception, School life, SES) on toothbrushing frequency in 2006, compared to 2010 and 2014".

The objectives of the present study were (1) to provide an update on the evolution of the tooth brushing frequency among French adolescents, and (2) to determine if the relationship between toothbrushing frequency and its socio-economic and behavioral associated factors changed over the years, using the data from the successive French HBSC studies of 2006, 2010 and 2014.

## Materials and methods

We used the data from the representative HBSC cross-sectional studies of 2006, 2010 and 2014 on French adolescents from 11 to 15 years old. Details on the survey can be found at http://www.hbsc.org.

Briefly, the HBSC studies are cross sectional studies, repeated every 4 years, describing adolescent's well-being, health behaviors and their social context. Our sample consists of adolescents from 11 to 15 years old selected using the international standardized HBSC sampling protocol. A two-stages cluster sampling over schools (public and private) and classes (two classes by school) was used. We used only the French cases included in the HBSC international database, which targeted adolescents of 11, 13 and 15 years old.

The full protocol of the HBSC studies has been described in previous articles and is designed to collect the most representative data and limit the risk of bias [18–20]. The questionnaire includes mandatory international questions, optional international thematic questions, and some specific national questions. It was validated on around 100 adolescents prior to the study, and double checked for translation from English. The minimal sample size (around 4500 for each study) was determined by the international HBSC protocol.

Information was collected over 2 months (22th April to 22th June in 2010) using an anonymous self-reported questionnaire. Data collectors were mostly teachers and nurses. The HBSC study received approval from the French data collection authority (CNIL) and the French ministry of education. Parents and students were given a consent form with the option to refuse before the adolescent's participation.

Tooth brushing frequency was assessed using the question "How often do you brush your teeth?" with five levels of response: More than once a day, Once a day, At least once a week but not daily, Less than once a week, Never. Data were pooled into two groups: "Tooth brushing at least twice a day" and "Tooth brushing once a day or less". This cutoff was chosen to match the international recommendations for tooth brushing and to avoid groups of subjects being too small. Toothbrushing frequency was our main outcome; other variables were considered as potential associated factors. Missing answers for this question led to exclusion from the analysis.

Despite being fairly constant over the years, the questionnaire has evolved slightly, especially as far as questions specific to the French version are concerned. Variables previously identified as associated with toothbrushing frequency and coming from items that remained similar in each study from 2006 to 2014 were considered for this paper (Table 1). They were divided in 4 main categories: Diet, Health and body, School life, Socio-economic status (SES).

The Body Mass Index (BMI) was used with Cole's age and sex specific cut-offs for overweight or obesity which allow the use of BMI for children. The Cantril score was used to measure overall life satisfaction and dichotomized at the usual threshold of 6 or over for good life satisfaction [21].

Analyses were performed using R [22] with the RStudio interface [23]. Chi square tests were used to assess the relationship between tooth brushing frequency and co variables.

**Table 1. Proportion of children brushing at least twice a day according to explanatory variables over the whole sample (bivariate analyses), significant variables at p<0.05 in bold.** N = 18727.

| Variable (% of the sample) | % | p-value |
|---|---|---|
| **DIET** | | |
| **Daily breakfast** | | **<0.0001** |
| Yes (58.4%) | 73.3 | |
| No (41.6%) | 67.3 | |
| **Daily fruit consumption** | | **<0.0001** |
| Yes (35.3%) | 77.4 | |
| No (64.7%) | 67.3 | |
| **Daily vegetable consumption** | | **<0.0001** |
| Yes (42.8%) | 74.5 | |
| No (57.2%) | 68.1 | |
| **Daily sweets consumption** | | **0.001** |
| Yes (25.1%) | 69.0 | |
| No (74.9%) | 71.5 | |
| **Daily soft drinks consumption** | | **<0.0001** |
| Yes (26.3%) | 67.3 | |
| No (73.7%) | 72.1 | |
| Being on a diet | | 0.14 |
| Yes (9.9% | 72.3 | |
| No (90.1%) | 70.6 | |
| **HEALTH AND BODY** | | |
| **Excellent self-reported health** | | **<0.0001** |
| Yes (35.9%) | 74.0 | |
| No (64.1%) | 69.0 | |
| **Overweight or obese** | | **<0.0001** |
| Yes (10.6%) | 59.6 | |
| No (89.4%) | 72.6 | |
| **Cantril score > = 6** | | **<0.0001** |
| Yes (84.1%) | 71.6 | |
| No (15.9%) | 66.6 | |
| **Body Image** | | **<0.0001** |
| About the right size (57.4%) | 72.5 | |
| Too thin (13.4%) | 69.0 | |
| Too fat (29.2%) | 68.2 | |
| Variable (% of the sample) | % | p-value |
| **SCHOOL LIFE** | | |
| **Academic delay** | | **<0.0001** |
| Yes (14.6%) | 65.4 | |
| No (85.4%) | 71.7 | |
| **Liking school** | | **<0.0001** |
| Yes (69.6%) | 72.0 | |
| No (30.4%) | 68.0 | |
| Stressed by school | | 0.06 |
| Yes (68.3%) | 70.4 | |
| No (31.7%) | 71.8 | |
| **Victim of violence at school** | | **<0.0001** |
| Yes (21.0%) | 66.0 | |

(*Continued*)

**Table 1.** (Continued)

| No (79.0% | 72.1 | |
|---|---|---|
| **Perceived school grades** | | <**0.0001** |
| Low (11.8%) | 66.1 | |
| Mid (36.8%) | 68.4 | |
| High (51.4%) | 73.8 | |
| **Support from classmates** | | <**0.0001** |
| Low (12.1%) | 66.6 | |
| Mid (54.6%) | 70.5 | |
| High (33.3%) | 73.0 | |
| School too demanding | | 0.28 |
| Low (36.2%) | 70.7 | |
| Mid (48.8%) | 71.2 | |
| High (15.0%) | 69.6 | |
| **Bullying others** | | <**0.0001** |
| Yes (36.0%) | 66.9 | |
| No (64.0% | 73.2 | |
| **Being bullied** | | <**0.0001** |
| Yes (32.8%) | 66.5 | |
| No (67.2%) | 73.0 | |
| Variable (% of the sample) | % | p-value |
| **SES** | | |
| **Perceived wealth** | | <**0.0001** |
| Low (7.4%) | 63.1 | |
| Mid (27.4%) | 67.6 | |
| High (65.2%) | 72.9 | |
| **Parental activity** | | <0.005 |
| Both parents working (75.1%) | 71.6 | |
| One parent working (20.8%) | 69.2 | |
| None working (3.1%) | 66.8 | |
| **Family Affluence Scale** | | <**0.0001** |
| Low (8.8%) | 66.1 | |
| Mid (35.5%) | 69.3 | |
| High (55.6%) | 72.8 | |

Three variables were used to assess SES: Perceived wealth, Parental employment and the Family Affluence Scale (FAS), a composite variable used in HBSC studies since 1990 to measure SES [24]. Due to very strong collinearity, we only used the most significant and discriminating variable in regression models: Perceived wealth.

Three regression models were used with toothbrushing frequency as the response variable to:

- Test for interactions between the year of study and each significant variable in order to assess any changes of the impact of these variables.

- Control for SES (measured by Perceived wealth) which was considered as the main potential causal factor associated with toothbrushing frequency while other groups of variables (Diet, Perceived health and body, school life) were considered as potential confounding factors (Table 2a, 2b and 2c).

**Table 2. Results from the three different logistic regressions on the whole sample with the adjusted odds-ratios (aOR) of brushing at least twice a day.** A. Dietary behavior. B. Health and body. C. School life.

| | aOR | p-value | CI $_{95\%}$ | |
|---|---|---|---|---|
| **A.** | | | | |
| Study year 2010 versus 2006 | 1.01 | 0.7393 | 0.94 | 1.09 |
| **Study year 2014 versus 2006** | **1.86** | **<0.0001** | 1.7 | 2.02 |
| Boy of 13 y/o versus 11 y/o boy | 1.11 | 0,0598 | 1 | 1.24 |
| Boy of 15 y/o versus 11 y/o boy | 1.18 | 0,0037 | 1.06 | 1.32 |
| **Girl of 11 y/o versus 11 y/o boy** | **1.51** | **<0.0001** | 1.35 | 1.69 |
| **Girl of 13 y/o versus 11 y/o boy** | **2.36** | **<0.0001** | 2.09 | 2.65 |
| **Girl of 15 y/o versus 11 y/o boy** | **2.71** | **<0.0001** | 2.39 | 3.07 |
| **Eating breakfast daily** | **1.38** | **<0.0001** | 1.29 | 1.48 |
| **Eating fruits daily** | **1.52** | **<0.0001** | 1.41 | 1.65 |
| **Eating vegetables daily** | **1.13** | **0.0010** | 1.05 | 1.22 |
| Eating sweets daily | 0.91 | 0.0194 | 0.83 | 0.98 |
| Drinking soft drinks daily | 0.91 | 0.0181 | 0.84 | 0.98 |
| **Wealth perceived: Mid vs Low** | **1.25** | **0.0008** | 1.1 | 1.43 |
| **Wealth perceived: High vs Low** | **1.66** | **<0.0001** | 1.47 | 1.89 |
| **B.** | | | | |
| Study year 2010 versus 2006 | 1.07 | 0.1041 | 0.99 | 1.16 |
| **Study year 2014 versus 2006** | **1.96** | **<0.0001** | 1.79 | 2.15 |
| Boy of 13 y/o versus 11 y/o boy | 1.07 | 0.2311 | 0.96 | 1.20 |
| Boy of 15 y/o versus 11 y/o boy | 1.04 | 0.4680 | 0.93 | 1.17 |
| **Girl of 11 y/o versus 11 y/o boy** | **1.57** | **<0.0001** | 1.39 | 1.77 |
| **Girl of 13 y/o versus 11 y/o boy** | **2.31** | **<0.0001** | 2.03 | 2.62 |
| **Girl of 15 y/o versus 11 y/o boy** | **2.54** | **<0.0001** | 2.23 | 2.89 |
| **Cantril $\geq$ 6** | **1.19** | **0.0007** | 1.08 | 1.31 |
| **BMI: Overweight or obese** | **0.65** | **<0.0001** | 0.58 | 0.73 |
| **Excellent perceived health** | **1.30** | **<0.0001** | 1.21 | 1.41 |
| Body image: too thin vs normal | 0.93 | 0.1880 | 0.84 | 1.04 |
| **Body image: too fat vs normal** | **0.88** | **0.0037** | 0.80 | 0.96 |
| Wealth perceived: Mid vs Low | 1.16 | 0.0456 | 1.00 | 1.33 |
| **Wealth perceived: High vs Low** | **1.47** | **<0.0001** | 1.29 | 1.69 |
| **C.** | | | | |
| Study year 2010 versus 2006 | 1.02 | 0.6111 | 0.94 | 1.11 |
| **Study year 2014 versus 2006** | **1.86** | **<0.0001** | 1.70 | 2.04 |
| Boy of 13 y/o versus 11 y/o boy | 1.15 | 0.0147 | 1.03 | 1.29 |
| Boy of 15 y/o versus 11 y/o boy | 1.18 | 0.0049 | 1.05 | 1.33 |
| **Girl of 11 y/o versus 11 y/o boy** | **1.56** | **<0.0001** | 1.39 | 1.76 |
| **Girl of 13 y/o versus 11 y/o boy** | **2.32** | **<0.0001** | 2.06 | 2.61 |
| **Girl of 15 y/o versus 11 y/o boy** | **2.62** | **<0.0001** | 2.31 | 2.97 |
| **Having academic delay** | **0.80** | **<0.0001** | 0.72 | 0.88 |
| Perceived school grades: Mid | 1.03 | 0.6189 | 0.92 | 1.15 |
| **Perceived school grades: High** | **1.24** | **0.0002** | 1.11 | 1.40 |
| Liking school | 1.02 | 0.6955 | 0.94 | 1.10 |
| Demand from school: Mid | 0.94 | 0.1313 | 0.87 | 1.02 |
| Demand from school: High | 0.90 | 0.0633 | 0.81 | 1.01 |
| **Classmate's support: Mid** | **1.17** | **0.0049** | 1.05 | 1.30 |
| **Classmate's support: High** | **1.29** | **<0.0001** | 1.15 | 1.45 |

*(Continued)*

**Table 2.** (Continued)

|  | aOR | p-value | CI $_{95\%}$ | |
|---|---|---|---|---|
| **Being bullied** | **0.84** | **<0.0001** | 0.78 | 0.90 |
| **Bullying others** | **0.86** | **<0.0001** | 0.80 | 0.93 |
| Wealth perceived: Mid vs Low | 1.19 | 0.0112 | 1.04 | 1.36 |
| **Wealth perceived: High vs Low** | **1.56** | **<0.0001** | 1.37 | 1.77 |

For variables with 3 levels, the "Low" level was used as reference. Due to the interaction, age and sex are described as one variable with 6 levels, the reference level being a boy of 11 y/o. Significant values at p<0.0001 in bold.
The reference level is a boy of 11 y/o in 2006. For example, on Table 2a, being a boy of 11 y/o in 2014 gives an aOR of 1.86 in favor of brushing at least twice a day versus a boy of 11 y/o in 2006. No significant interactions were identified, so the aOR of brushing at least twice a day for a 15 y/o girl in 2014 versus a 15 y/o girl in 2006 is not significantly different from 1.86.

Three final models were obtained, each including the Year of the study, Age, Sex, Perceived wealth, and the significant (p<0.001) variables describing either Diet (Table 2a), Health and body (Table 2b) or School life (Table 2c), as well as significant interactions if any. The impacts of variables were measured using adjusted odds ratio (aOR).

Interactions were tested between age, sex and the other significant variables, since age and sex are commonly involved in interactions regarding health behaviors.

As is often done in HBSC analyses, we used a significance level α of p<0.001.

## Results

From the 29226 questionnaires of the 2006, 2010 and 2014 studies, keeping only the questionnaires of students of 11, 13 and 15 years old who were included in the international HBSC studies (10309 exclusions) and who answered the question about toothbrushing (352 exclusions), 18727 questionnaires were analyzed, 7135 for 2006, 6103 for 2010 and 5489 for 2014 (Fig 1).

The sample was almost perfectly balanced, with 9352 boys (49.9% overall, 49.7% in 2006, 50.1% in 2010 and 2014) and 9375 girls. 6294 students were 11 years old (33.6%), 6609 were 14 (35.3%), and 5824 were 15 (31.1%) (Fig 2).

Toothbrushing frequency improved significantly in 2014 compared to 2010 and 2006. While in 2006, 66.8% of students reported brushing twice a day or more, their percentage was 68.0% in 2010 (not significantly different from 2006) and 78.8% in 2014 (significantly different from 2006 and 2010, p<0.0001).

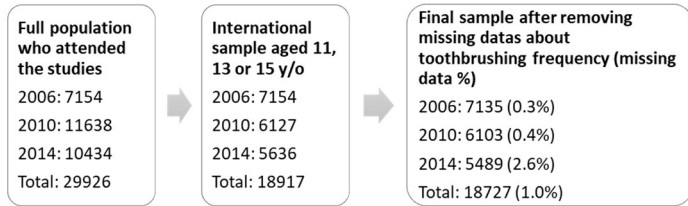

**Fig 1. Flow chart.** In 2006, the study was designed to comply with the international requirements while, in 2010 and 2014, the sample was larger to better describe a specific French population.

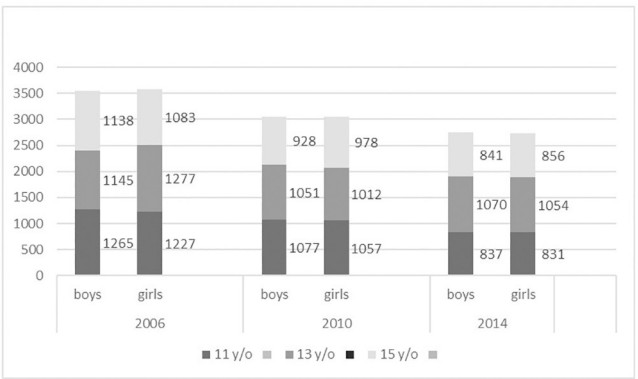

**Fig 2. Population according to age, sex and year of study.**

Age and sex are strong factors associated with tooth brushing frequency, and their impact changed little over the years (Fig 3). Brushing frequency among boys is not associated with their age, while toothbrushing frequency increased among girls as they grow older. However, this improvement with age among girls tended to diminish in the 2014 study due to a faster increase of the twice a day brushing among younger girls (+11.4% from 2010 to 2014 for the 11 year old girls) than among older girls (+9.3% over the same period for the 15 year old girls). Being a girl (and, even more, an older girl) remained the most important associated factor over the 3 studies with around 10% more girls than boys brushing twice a day.

Among the whole sample (2006, 2010 and 2014 grouped together), almost all studied variables were associated with tooth brushing frequency (Table 1).

In particular, healthy eating habits (daily consumption of fruit, vegetables or breakfast) were associated with an increased proportion of twice a day tooth brushing among students; up to a 10% increase for fruit consumption (Table 1). High socio-economic status markers (Family affluence scale, perceived family wealth, parental employment), having no problem in school life (no involvement in bullying or violence, not lagging behind academically, liking school), excellent self-reported health, good body perception, good life satisfaction and not

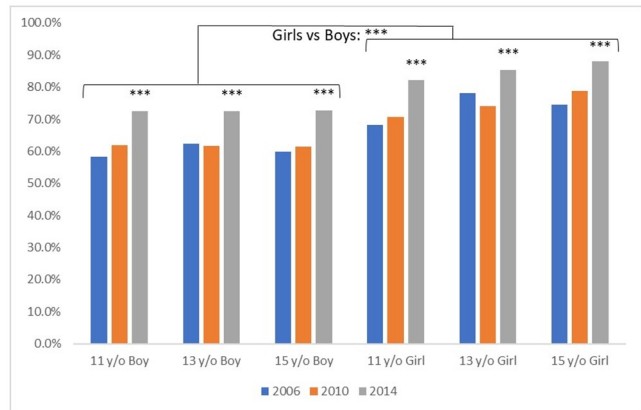

**Fig 3. Proportion of adolescents brushing at least twice a day according to age, sex and year of the study.** The proportion is significantly higher (***: p>0.0001) among girls than boys, and in 2014 compared to 2010 and 2006 in every age/sex group (***). A slightly increased toothbrushing frequency among older girls (13 y/o vs 11 y/o, and 15 y/o vs 13 and 11 y/o) can also be seen.

being overweight or obese were also associated with a higher toothbrushing frequency (Table 1).

The vast majority of the missing values in regression models were due to the BMI item (2074 missing values). Among the adolescents with a missing BMI, 67.2% brushed twice a day. Since this value is between the "Normal weight" (72.6% of them brush twice a day) and the "Overweight or obesity" (59.6%) groups, we assumed that those missing value were from adolescents with normal weight as much as overweight or obese adolescents. The most probable reason for those missing value is adolescents not knowing their height or weight. Those missing values were considered as missing at random.

Results from the regressions (Table 2a, 2b and 2c) confirmed the findings on the impact of age and sex, including an interaction term. No significant effect of age on toothbrushing frequency among boys was identified, while there was a slight significant effect among girls. Adjusted OR (aOR) at 11 y/o ranged from 1.40 to 1.57 in the models, and from 2.54 to 2.71 at 15 y/o. Results shown are the cumulative effect of being a girl and being 11, 13 or 15 y/o, the base level being an 11 y/o boy. In each model, being a girl (especially an older girl) was the strongest factor associated with an appropriate toothbrushing frequency. The diminution over the studies of the impact of age on girls' toothbrushing frequency, identified in bivariate analysis, was not strong enough to be kept in the regression model.

The year 2014 was the second strongest associated factor and no interaction between the year of the study and any other variable associated with tooth brushing frequency was found. All associated factors stayed almost constant over the years.

SES, measured via perceived wealth, was the $3^{rd}$ strongest associated factor.

In the "Dietary behavior" model (Table 2a), having a breakfast (aOR 1.38, $CI_{95\%}$ [1.29–1.48]) and eating fruits every day (aOR 1.52, $CI_{95\%}$ [1.41–1.64]) were the strongest positive factors associated with twice a day tooth brushing.

In the "Health and body" model (Table 2b), being overweight (aOR 0.65, $CI_{95\%}$ [0.58–0.73]) was a strong negative factor associated with toothbrushing frequency, while an excellent perceived health (aOR 1.30, $CI_{95\%}$ [1.21–1.41]) was the strongest positive associated factor.

In the "School life" model (Table 2c), Classmate's support and high perceived school grades were positive associated factors, while having academic delay or being part of bullying as a bully or a victim were unfavorable factors associated with twice a day toothbrushing.

## Discussion

We observed an increase of twice a day tooth brushing among adolescents from around 70% in 2006 and 2010 to 80% in 2014. It is likely that knowledge about the appropriate toothbrushing frequency led to a response (social desirability) bias overestimating the real toothbrushing frequency. It is however likely that this bias stayed constant over the 3 studies as no important changes were made to the protocol of the studies, and thus that the increase of twice a day toothbrushing reflect a real trend. Societal pressure may have increased, but would be more likely to be associated with an increase in missing answers to the question in 2014, than to false declaration to this extent. The national voluntary prevention program mentioned in introduction was updated to be more efficient on the same period, however it still only reaches one third of the targeted children and likely can't be the only cause of the changes we observed. A detailed investigation on the French national and regional prevention programs as well as societal changes could help understand this evolution.

The strongest factor explaining differences in toothbrushing behavior is being a girl, which has already been known for a long time [14]. A high SES was a strong factor associated with adequate toothbrushing in each model.

A low tooth brushing frequency is associated with poor perceived health, unhealthy eating habits and overweight or obesity, all of these elements being likely consequences of unhealthy lifestyles and poor health knowledge. This should be a major concern since the association of these unfavorable factors leads to an increased risk of oral and general health diseases. Difficulties in school life and low perceived family wealth are often associated with low SES, which is often found as a socio-environmental factor associated with poor oral hygiene. Our findings match previous results [17].

An adequate toothbrushing frequency comes from the combination of knowledge (Health literacy) and motivation [25]. Age, gender and SES are most likely associated with both elements and possibly causal factors for toothbrushing frequency. On the other hand, the "Health and body" and "Dietary behavior" variables are consequences of health literacy and should be considered as confounding factors considering toothbrushing frequency. They are however of interest due to the cumulative effect they may have with toothbrushing frequency in preventing or facilitating oral diseases.

Variables describing "School life" are harder to interpret. They may be consequences of SES (a low SES is associated with lower academic results and a higher risk of bullying), and thus confounding factors, but could also be causal factors for knowledge (better academic results linked to a better health literacy) and motivation (via better relationships with classmates). Those two interpretations are likely coexisting.

The improvement in tooth brushing frequency is shared among the whole population with no exception. All associated factors stayed constant in 2006, 2010 and 2014. Traditional populations showing a high risk of low oral hygiene (boys, populations with low social and economic status, cumulating other risk factors such as unhealthy lifestyles) benefited from this change as well and were not left behind. Although they still have more room for improvement than the other adolescents, this shows that there are no inevitable consequences for those populations, which are often considered to be too hard to reach for prevention.

Most missing values were linked to BMI. Adolescents may not have known their height or weight, or did not want to disclose it. A lack of knowledge seems more likely since both variables were missing at roughly the same ratio, and while weight is a sensitive question, height isn't. This is backed by the fact that their brushing frequency is between the frequencies of the "overweight" group and the "normal weight" group, advocating for a mixed group.

Global population-based prevention programs should be maintained to address the broadest proportion of children and adolescents. Some high-risk groups should be targeted with more specific programs, such as multidisciplinary counselling or therapeutic patient education to improve their knowledge and behavior considering oral health [12, 26].

Cohort studies are needed to confirm the causality hypothesis we developed in discussion, which cross-sectional studies like HBSC can't. While this sample is highly representative of the French population, it would be interesting to check if the associations identified during this study are similar in other countries, which could be done using the result from other HBSC studies.

## Conclusion

Tooth brushing frequency increased among adolescents from 2010 to 2014, probably due to an efficient improvement of the prevention programs over recent decades. Although socio-economic inequalities in oral hygiene habits persists, populations at risk still benefitted from the improvement in toothbrushing frequency like the rest of the population.

Oral hygiene stays closely associated with other healthy behaviors and should be considered as part of a whole when addressing health education. It is possible to improve oral hygiene and

while the whole population could benefit from it, some populations are more in need and could use oral hygiene and diet improvement to reduce their risk for oral diseases.

Comparison with the other countries undertaking the HBSC study could now offer a broader vision of oral hygiene in western countries.

## Supporting information

**S1 Checklist.**
(DOC)

**S1 Questionnaire.**
(PDF)

## Acknowledgments

Health Behaviour in School-Aged Children (HBSC) is an international study carried out in collaboration with World Health Organization/Europe WHO/EURO. The French National Coordinator is Emmanuelle Godeau.

## Author Contributions

**Conceptualization:** Emmanuelle Godeau.

**Data curation:** Emmanuelle Godeau.

**Formal analysis:** Gabriel Fernandez de Grado.

**Methodology:** Gabriel Fernandez de Grado, Virginie Ehlinger.

**Project administration:** Emmanuelle Godeau, Catherine Arnaud.

**Resources:** Catherine Arnaud, Nadia Benkirane-Jessel.

**Software:** Catherine Arnaud.

**Supervision:** Cathy Nabet, Nadia Benkirane-Jessel, Anne-Marie Musset, Damien Offner.

**Writing – original draft:** Gabriel Fernandez de Grado.

**Writing – review & editing:** Gabriel Fernandez de Grado, Virginie Ehlinger, Emmanuelle Godeau, Catherine Arnaud, Cathy Nabet, Nadia Benkirane-Jessel, Anne-Marie Musset, Damien Offner.

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
