## [Decision Letter · Decision Letter 0]

17 Dec 2020

PONE-D-20-30886

Changes in toothbrushing frequency and its associated factors from 2006 to 2014 among French adolescents

PLOS ONE

Dear Dr. Fernandez de Grado,

Thank you for submitting your manuscript to PLOS ONE. After careful consideration, we feel that it has merit but does not fully meet PLOS ONE’s publication criteria as it currently stands. Therefore, we invite you to submit a revised version of the manuscript that addresses the points raised during the review process.

We look forward to receiving your revised manuscript.

Kind regards,

Frédéric Denis, Ph.D.

Academic Editor

PLOS ONE

Journal Requirements:

2.) Please provide additional details regarding participant consent. In the ethics statement in the Methods and online submission information, please ensure that you have specified what type you obtained (for instance, written or verbal, and if verbal, how it was documented and witnessed). If your study included minors, state whether you obtained consent from parents or guardians. If the need for consent was waived by the ethics committee, please include this information.

3.) We note that you have indicated that data from this study are available upon request. PLOS only allows data to be available upon request if there are legal or ethical restrictions on sharing data publicly. For information on unacceptable data access restrictions, please see http://journals.plos.org/plosone/s/data-availability#loc-unacceptable-data-access-restrictions.

Reviewers' comments:

Reviewer's Responses to Questions

**Comments to the Author**

1. Is the manuscript technically sound, and do the data support the conclusions?

Reviewer #1: Yes

Reviewer #2: No

Reviewer #3: Partly

2. Has the statistical analysis been performed appropriately and rigorously? 

Reviewer #1: I Don't Know

Reviewer #2: No

Reviewer #3: N/A

3. Have the authors made all data underlying the findings in their manuscript fully available?

Reviewer #1: No

Reviewer #2: No

Reviewer #3: Yes

4. Is the manuscript presented in an intelligible fashion and written in standard English?

Reviewer #1: No

Reviewer #2: Yes

Reviewer #3: Yes

5. Review Comments to the Author

Reviewer #1: Thank you very much for submitting your research to Plos One. The reviewer would like to make several comments on your article.

1.What is the main message the authors would like to deliver?

2.Pleaes comment on the accuracy of the self-reported questionnaire.

3.Please provide the full paragraph of self-reported questionnaires.

4.Please comment on this.

This cutoff was chosen to match the international recommendations for tooth brushing and to avoid groups of subjects being too small.

5.Please comment on confounding factor.

6.Can you provide the data for more recent years.

7.Can the individual be followed up.

8.What is the rationale for toothbrushing frequency?

9.Please provide the possible explanation.

10. Please provide the clinical relevance.

Thank you very much.

Reviewer #2: The paper shows that among French adolescents, toothbrushing frequency changed from 2006 to 2014 using the data of Health Behavior in School-aged Children (HBSC) survey.

This is an interesting study. However, I would like to make some points regarding the manuscript. The article needs to be revised. First, the authors should follow the STROBE guideline if this study is an observational study. Second, the logic is unclear. The authors answered to a reviewer in the last revision as below.

However, the current form is not appropriate.

1. I actually wasn’t entirely sure what the research question was. I believe the question was to determine whether the percentage adolescents who brushed their teeth twice a day changed over time. What was confusing was that the predictors were all individual/family level variables. So there seemed to be a mismatch between the question and the factors examined. To determine predictors of change over time, I would look to broader social factors. The authors seemed to recognize this, because in their discussion, they speculated about the role of a social program on the outcome. In their study, however, they did not examine the effect of this social program analytically. Thus, there is a mismatch between the research question and the predictors included in the study. To me, this is a fatal flaw of the paper.

Answer: It seems our secondary objective was misunderstood by this reviewer which led to many criticisms. We wanted to determine if the relationship between toothbrushing frequency and its predictors changed over time among French adolescents. We did NOT aim to determine predictors of changes in toothbrushing frequency; a goal that would require a longitudinal study. We did NOT aim to offer an analytical model. We reformulated our objectives : “The objectives of the present study were (1) to provide an update on the evolution of the tooth brushing frequency among French adolescents, and (2) to determine if the relationship between toothbrushing frequency and its socio-economic and behavioral predictors changed over the years, using the data from the successive French HBSC studies of 2006, 2010 and 2014.”

2. The authors seemed to have ideas about the importance of different classes of predictors, but they failed to describe any theoretical underpinning for their ideas. I believe we have passed the time when atheoretical research is acceptable. At this time, the best way for research to make a contribution is if it is theory-driven. The authors should ground their work in theory. Doing so would help the authors address the point above. This is a flaw that is addressable, but addressing it would turn this into a new paper.

Answer: We indeed have theory about the roles of the identified predictors, which are discussed in the paper. We rewrote the discussion to make it clearer. However, we insist on the fact that this article’s main goal was to identify changes in adolescent toothbrushing, and is thus a descriptive article whose goal is to offer information about the situation, not to offer a full theory on the relationships between toothbrushing and all its associated factors.

3. The analytic strategy is weak. There are serious drawbacks to using stepwise regression, one being that results using this approach are not generalizable. The results are contingent on the specific sample on which the analysis is run. See: https://www.stata.com/support/faqs/statistics/stepwiseregression-problems/ This is a flaw that is addressable, but addressing it would turn this into a new paper. My advice would be to start with theory and figure out the ways in which their study can help inform our understanding of that theory.

Answer: We rewrote the article, deleted the part about stepwise regression and choose to present only descriptive models to make it clearer that this is a descriptive article. We never aimed to offer more than a description of toothbrushing frequency and its associated factors among French adolescents. Our discussion mentions the possible consequences and origin of these associations, but we did not aim to offer a full theory about the mechanisms leading to these associations.

Furthermore, to be accepted for publication in PLOS ONE, research articles must satisfy the following criteria:

1. The study presents the results of original research.

2. Results reported have not been published elsewhere.

3. Experiments, statistics, and other analyses are performed to a high technical standard and are described in sufficient detail.

4. Conclusions are presented in an appropriate fashion and are supported by the data.

5. The article is presented in an intelligible fashion and is written in standard English.

6. The research meets all applicable standards for the ethics of experimentation and research integrity.

7. The article adheres to appropriate reporting guidelines and community standards for data availability.

The current form is not acceptable in this journal policy.

TITLE

1) Please add the study design following the STROBE guideline if it is a cross-sectional study.

INTRODUCTION

1) The logic is unclear. If this is a descriptive epidemiology, the authors should need hypothesis formulation. The structure of this paper should be changed dramatically. If it’s a cross-sectional study, they need to add hypothesis before aim. The PECO model should be considered.

2) The topic of dental caries occupies most of introduction section. The authors should make it short.

MATERIALS & METHODS

1) Please add some comments following the STROBE checklist.

2) Children or adolescents? Please unify the expression.

3) The authors stated that “Details on the survey can be found at http://www.hbsc.org.” and “The full protocol of the HBSC studies has been described in previous articles (17-19).” However, the authors should add more details for readers following the STROBE checklist. Furthermore, I could not find any supplemental files. 4) If the authors would like to compare toothbrushing frequency in 2010 and 2014 against 2006, and to investigate whether a temporal association between toothbrushing frequency and its related factors changed over time, the statistical analyses should be changed. The logistic regression models should be revised based on the hypothesis/PECO model.

RESULTS

1) Please add the characteristic table following the STROBE guideline.

2) Please the results based on new methods.

DISCUSSION

1) The data will be changed by new results.

2) The authors cannot use “predictive” and “improve” because this is a cross-sectional study and not cohort study.

3) What is the sentence, “An adequate toothbrushing frequency is the fruit of knowledge (Health literacy) and motivation”? Please revise it appropriately.

4) In a paragraph, there is only one sentence, “All those results match previous results (16).” The authors should revise the part.

5) Please add some comments about limitations, such as a cross-sectional study, no data of important confounders, no generalizability, bias, etc.

6) Please revise the conclusions because this is a cross-sectional study.

Reviewer #3: - it is based on questionnaire data (survey) that may over or under estimate the real tooth brush frequency.

- If the authors presume the incresced toothbrushing frequency in 2014 is related to the educational program, I recommend to mention the subjective evidence uch as statistical analysis.

6. PLOS authors have the option to publish the peer review history of their article (what does this mean?). If published, this will include your full peer review and any attached files.

Reviewer #1: No

Reviewer #2: No

Reviewer #3: No

---

## [Author Response · Author response to Decision Letter 0]

9 Feb 2021

All comments from reviewers are answered in the response file. Additionally, here are our answers to the Journal requirements:

1) Concerning style requirements, we updated the organization of the article, the naming of figures and the organization of tables according to PLOS ONE's style requirements. We used page break to keep tables organized since they are quite long.

2) Concerning consent, we corrected the sentence to include more details: "Parents and students were given a consent form with the option to refuse before the adolescent’s participation."

3) Concerning data availability, we realized that the data are now old enough to be on public access, and we gave the link to the databases, both on editorial manager and in the manuscript: "Data are fully available on: https://www.uib.no/en/hbscdata/113290/open-access"

---

## [Decision Letter · Decision Letter 1]

4 Mar 2021

PONE-D-20-30886R1

Changes in tooth brushing frequency and its associated factors from 2006 to 2014 among French adolescents: results from three repeated cross sectional HBSC studies

PLOS ONE

Dear Dr. Fernandez de Grado,

Thank you for submitting your manuscript to PLOS ONE. After careful consideration, we feel that it has merit but does not fully meet PLOS ONE’s publication criteria as it currently stands. Therefore, we invite you to submit a revised version of the manuscript that addresses the points raised during the review process.

We look forward to receiving your revised manuscript.

Kind regards,

Frédéric Denis, Ph.D.

Academic Editor

PLOS ONE

Journal Requirements:

Reviewers' comments:

Reviewer's Responses to Questions

**Comments to the Author**

1. If the authors have adequately addressed your comments raised in a previous round of review and you feel that this manuscript is now acceptable for publication, you may indicate that here to bypass the “Comments to the Author” section, enter your conflict of interest statement in the “Confidential to Editor” section, and submit your "Accept" recommendation.

Reviewer #1: All comments have been addressed

Reviewer #2: (No Response)

2. Is the manuscript technically sound, and do the data support the conclusions?

Reviewer #1: Yes

Reviewer #2: No

3. Has the statistical analysis been performed appropriately and rigorously? 

Reviewer #1: Yes

Reviewer #2: No

4. Have the authors made all data underlying the findings in their manuscript fully available?

Reviewer #1: Yes

Reviewer #2: No

5. Is the manuscript presented in an intelligible fashion and written in standard English?

Reviewer #1: Yes

Reviewer #2: Yes

6. Review Comments to the Author

Reviewer #1: Dear Authors

I extent my sincere thanks for submitting your revised manuscript for the further review. Authors have answered all the queries very nicely. This paper is of excellent merit and it seems fully acceptable for publication in Medicine.

Thank you very much

Reviewer #2: The paper was overall improved. However, there are some issues. The logic is still unclear.

1) The authors answered;

‘Our secondary question was “Among adolescents, what is the effect of associated factors (Diet, Health and body perception, School life, SES) in 2006, compared to 2010 and 2014 on toothbrushing frequency”’.

However, the hypothesis is inappropriate, because the participants were not same. We can only investigate the association between tooth brushing frequency in 2006 and related factors in 2006 but not 2010/2014.

2) The title of table 2 is unclear. Please add more detail comments. What is the odds ratio for? Furthermore, if the dependent value is “Tooth brushing at least twice a day” vs. “Tooth brushing once a day or less” in 2014, the independent values should be the data in 2014 but not 2006/2010. The analyses are inappropriate, even though “all predictors stayed almost constant over the years”. The authors can’t separate the table 2 such as 2a, 2b and 2c, too.

3) What is the mechanism? Do the authors mean that eating breakfast daily, eating fruits daily, eating vegetables daily, wealth perceived, BMI, excellent perceived health, body image, having academic delay, perceived school grades, classmate’s support, being bullied, bullying others is associated with TBF? Are these factors risk for decreasing TBF? If yes, they should show the appropriate references in the introduction and discuss the mechanisms. If no, the logistic regression analyses should be deleted or revised.

4) The authors can’t use the words, “predictors” and “predictive factors”, because this is a cross-sectional study but not a cohort study. If the new results are similar after re-analyses, they may change the conclusion; “Among French adolescents, TBF improved from 2006 to 2014. The TBF was significantly associated with some health behaviors in each year.”

7. PLOS authors have the option to publish the peer review history of their article (what does this mean?). If published, this will include your full peer review and any attached files.

Reviewer #1: **Yes: **Jun-Beom Park

Reviewer #2: No

---

## [Author Response · Author response to Decision Letter 1]

10 Mar 2021

Answer to the reviewers

Dear Reviewers, 

Thank you for your constructive comments. You will find in this text, in red, all the changes we have made (re-written sentences, corrections, and adjunctions), following your remarks and advices. Changes in the article are in revision mode.

Please find below our answers concerning some of your questions and points:

1) The authors answered;

‘Our secondary question was “Among adolescents, what is the effect of associated factors (Diet, Health and body perception, School life, SES) in 2006, compared to 2010 and 2014 on toothbrushing frequency”’.

However, the hypothesis is inappropriate, because the participants were not same. We can only investigate the association between tooth brushing frequency in 2006 and related factors in 2006 but not 2010/2014.

Of course, in our study we compared the association between TBF in 2006 and its related factors in 2006, with the association of TBF in 2010 and its related factors in 2010 (and the same in 2014). We did not compare related factors of 2010 or 2014 with TBF in 2006 which would make no sense.

We rewrote the sentence to make this clearer.

“Among adolescents, what is the effect of associated factors (Diet, Health and body perception, School life, SES) on toothbrushing frequency in 2006, compared to this association in 2010 and 2014 “

Actually, when studying a specific age group (children from 11 to 15 in our study), it is impossible to keep the same sample over years. However, using successive representative samples, it is possible to compare those samples and then describe an evolution over the years.

2) The title of table 2 is unclear. Please add more detail comments. What is the odds ratio for? Furthermore, if the dependent value is “Tooth brushing at least twice a day” vs. “Tooth brushing once a day or less” in 2014, the independent values should be the data in 2014 but not 2006/2010. The analyses are inappropriate, even though “all predictors stayed almost constant over the years”. The authors can’t separate the table 2 such as 2a, 2b and 2c, too.

We added more details to this table’s title which was unclear. The new title better explains the analyses that were done, without the need to refer to the manuscript’s text.

“Tables 2: Results from the three different logistic regressions on the whole sample with the adjusted odd-ratios of brushing at least twice a day: The reference level is a boy of 11 y/o in 2006. For example, on table 2a, being a boy of 11 y/o in 2014 gives an aOR of 1.86 in favor of brushing at least twice a day. No significant interactions were identified, so the aOR of brushing at least twice a day for a 15 y/o girl in 2014 versus a 15 y/o girl in 2006 is not significantly different from 1.86.”

3) What is the mechanism? Do the authors mean that eating breakfast daily, eating fruits daily, eating vegetables daily, wealth perceived, BMI, excellent perceived health, body image, having academic delay, perceived school grades, classmate’s support, being bullied, bullying others is associated with TBF? Are these factors risk for decreasing TBF? If yes, they should show the appropriate references in the introduction and discuss the mechanisms. If no, the logistic regression analyses should be deleted or revised.

We discuss those elements in discussion, stating that most of the elements associated with toothbrushing frequency are probably confounding factors sharing common causal factors (SES, health knowledge, lifestyle) with toothbrushing frequency. This is however of interest since some of these factors are also predictors of oral health, just like TBF. Of course, those are only hypothesis in discussion since we have no way to determine causality links. 

“A low tooth brushing frequency is associated with poor perceived health, unhealthy eating habits and overweight or obesity, all of these elements being likely consequences of unhealthy lifestyles and poor health knowledge. This should be a major concern since the association of these unfavorable factors leads to an increased risk of oral and general health diseases. Difficulties in school life and low perceived family wealth are often associated with low SES, which is often found as a socio-environmental predictor for poor oral hygiene. Our findings match previous results (17).

An adequate toothbrushing frequency comes from the combination of knowledge (Health literacy) and motivation (25). Age, gender and SES are most likely predictors of both elements and possibly causal factors for toothbrushing frequency. On the other hand, the “Health and body” and “Dietary behavior” variables are consequences of health literacy and should be considered as confounding factors considering toothbrushing frequency. They are however of interest due to the cumulative effect they may have with toothbrushing frequency in preventing or facilitating oral diseases.

Variables describing “School life” are harder to interpret. They may be consequences of SES (a low SES is a predictor of lower academic results and a higher risk of bullying), and thus confounding factors, but could also be causal factors for knowledge (better academic results linked to a better health literacy) and motivation (via better relationships with classmates). Those two interpretations are likely coexisting.”

4) The authors can’t use the words, “predictors” and “predictive factors”, because this is a cross-sectional study but not a cohort study. If the new results are similar after re-analyses, they may change the conclusion; “Among French adolescents, TBF improved from 2006 to 2014. The TBF was significantly associated with some health behaviors in each year.”

This term still seems confusing despite our explanation in our previous answer that we used “predictor” with the statistical meaning of “associated factor” with no causality implication. Therefore, and to make it clearer to the readers, we replaced every instance of “predictor” and “predictive factor” by “associated factor” or “factor associated with…”. 

Following your suggestion, we changed our conclusion in the abstract to “Among French adolescents, TBF improved from 2006 to 2014. TBF was significantly associated with other health behaviors. These associations stayed similar in 2006, 2010 and 2014.”

We really hope that these answers and corrections will meet your expectations.

Best regards, 

The authors.

---

## [Editor Report · Decision Letter 2]

12 Mar 2021

Changes in tooth brushing frequency and its associated factors from 2006 to 2014 among French adolescents: results from three repeated cross sectional HBSC studies

PONE-D-20-30886R2

Dear Dr. Fernandez de Grado,

We’re pleased to inform you that your manuscript has been judged scientifically suitable for publication and will be formally accepted for publication once it meets all outstanding technical requirements.

Kind regards,

Frédéric Denis, Ph.D.

Academic Editor

PLOS ONE
---

## [Editor Report · Acceptance letter]

17 Mar 2021

PONE-D-20-30886R2 

Changes in tooth brushing frequency and its associated factors from 2006 to 2014 among French adolescents: results from three repeated cross sectional HBSC studies 

Dear Dr. Fernandez de Grado:

I'm pleased to inform you that your manuscript has been deemed suitable for publication in PLOS ONE. Congratulations! Your manuscript is now with our production department. 

Kind regards, 

on behalf of

Dr. Frédéric Denis 

Academic Editor

PLOS ONE